# Elevated Serum Urea-to-Creatinine Ratio and In-Hospital Death in Patients with Hyponatremia Hospitalized for COVID-19

**DOI:** 10.3390/biomedicines11061555

**Published:** 2023-05-27

**Authors:** Giuseppe Regolisti, Paola Rebora, Giuseppe Occhino, Giulia Lieti, Giulio Molon, Alessandro Maloberti, Michela Algeri, Cristina Giannattasio, Maria Grazia Valsecchi, Simonetta Genovesi

**Affiliations:** 1Clinica e Immunologia Medica, Azienda Ospedaliero-Universitaria di Parma, 43100 Parma, Italy; giuseppe.regolisti@unipr.it; 2Department of Medicine and Surgery, University of Parma, 43126 Parma, Italy; 3Bicocca Bioinformatics, Biostatistics and Bioimaging Centre-B4, School of Medicine and Surgery, Milano-Bicocca University, 20126 Milan, Italy; paola.rebora@unimib.it (P.R.); giuseppe.occhino@unimib.it (G.O.); grazia.valsecchi@unimib.it (M.G.V.); 4School of Medicine and Surgery, Milano-Bicocca University, 20126 Milan, Italy; g.lieti@campus.unimib.it (G.L.); alessandro.maloberti@unimib.it (A.M.); cristina.giannattasio@unimib.it (C.G.); 5Cardiology Department, Istituto Ricovero Cura Carattere Scientifico (IRCCS) Sacro Cuore Don Calabria Hospital, Negrar di Valpolicella, 37024 Verona, Italy; giulio.molon@sacrocuore.it; 6Cardiology 4, Cardio Center, ASST-GOM Niguarda, Niguarda Hospital, 20162 Milan, Italy; michela.algeri@ospedaleniguarda.it; 7Istituto Auxologico Italiano, Istituto Ricovero Cura Carattere Scientifico (IRCCS), 20135 Milan, Italy

**Keywords:** sodium, hyponatremia, urea-to-creatinine ratio, COVID-19, mortality, intensive care unit

## Abstract

Hyponatremia is associated with adverse outcomes in hospitalized patients. An elevated value of the serum urea-to-creatinine ratio (UCR) has been proposed as a proxy of hypovolemia. The aim of this study was to investigate the relationship between the UCR and in-hospital death in patients hospitalized with COVID-19 and hyponatremia. We studied 258 patients admitted for COVID-19 between January 2020 and May 2021 with serum sodium at < 135 mmol/L. The primary end-point was all-cause mortality. A 5-unit increase in the serum UCR during hospital stays was associated with an 8% increase in the hazard of all-cause death (HR = 1.08, 95% CI: 1.03–1.14, *p* = 0.001) after adjusting for potential confounders. In patients with a UCR > 40 at baseline, a > 10 mmol/L increase in serum sodium values within the first week of hospitalization was associated with higher odds of in-hospital death (OR = 2.93, 95% CI: 1.03–8.36, *p* = 0.044) compared to patients who experienced a < 10 mmol/L change. This was not observed in patients with a UCR < 40. Hypovolemia developing during hospital stays in COVID-19 patients with hyponatremia detected at hospital admission bears an adverse prognostic impact. Moreover, in hypovolemic patients, a > 10 mmol/L increase in serum sodium within the first week of hospital stays may further worsen the in-hospital prognosis.

## 1. Introduction

Hyponatremia is the most frequent electrolyte disorder in hospitalized patients and is associated with adverse outcomes [1,2]. Specifically, in patients with bacterial pneumonia, hyponatremia can be associated with increased mortality [3,4] and an increased risk of admission to the intensive care unit (ICU) [4].

Recently, the prognostic impact of hyponatremia has also been investigated in patients with COVID-19 hospitalized for interstitial pneumonia. Several pathophysiological and/or pharmacological mechanisms may explain hyponatremia in patients with COVID-19 and interstitial pneumonia. Inappropriate antidiuretic hormone (ADH) secretion is frequently involved. In fact, high circulating interleukin-6 (IL-6) levels may promote increased ADH secretion, as IL-6 can cross the blood–brain barrier and directly stimulate hormone release by supraoptic and paraventricular nuclei [5]. An inverse relationship between circulating levels of IL-6 and serum sodium concentration has been reported previously [6]. Moreover, increased ADH secretion may occur in COVID-19 patients due to hypovolemia, which can develop because of sodium loss during profuse diarrhea and vomiting. In addition, sodium loss during prolonged diuretic treatment with concomitant administration of hypotonic fluids (e.g., solutions for parenteral nutrition) in fluid-overloaded patients may represent another important cause of hyponatremia. Finally, because the entry of SARS-CoV-2 into the host cell via ACE2 disrupts the renin–angiotensin–aldosterone system (RAAS), creating an imbalance between ACE and ACE2 with an increased inflammatory response, we may speculate that this mechanism may also contribute to the development of hyponatremia in COVID-19 patients [7]. In this population, hyponatremic patients have been reported to have an increased risk of death in some [8,9,10] but not all [11,12] studies.

While hyponatremia can be associated with normal, decreased, or increased extracellular fluid volume [13], the clinical diagnosis of volume depletion is challenging, even upon expert judgment [14,15]. An elevated (≥20) value of blood urea nitrogen (BUN)-to-serum-creatinine (sCr) (BUN/sCr) ratio has been proposed as a clinically useful proxy of hypovolemia independent of serum sodium concentration [16,17]. In fact, it is expected that a low effective arterial blood volume (EABV) will elicit a baroreceptor-mediated neurohormonal response, resulting in increased sodium and water reabsorption in the kidney proximal tubule and decreased tubular flow in the distal nephron, thus enhancing flow-dependent urea reabsorption in the collecting duct [18].

Indeed, an elevated BUN/sCr ratio value predicts adverse outcomes both in the general population [19] and in different clinical scenarios, including congestive heart failure [20,21], acute myocardial infarction [22,23], ischemic stroke [24,25,26], and subarachnoid hemorrhage [27].

In a recent study conducted in a cohort of Chinese patients hospitalized with COVID-19, an elevated BUN/sCr ratio value was identified as a potentially useful prognostic index [28]. Interestingly, Tzoulis et al. suggested that COVID-19 patients with hyponatremia and concomitant volume depletion, as judged based on a serum urea concentration value > 5 mmol/L, may be at a higher risk of death [11]. In Europe, as opposed to the US, urea is usually assayed in lieu of BUN. Based on the molecular weight expressed in daltons, serum urea concentration is approximately equivalent to half of the BUN value (i.e., 28/60 or 0.446) [29].

On these grounds, we decided to investigate whether an elevated serum urea-to-creatinine ratio (UCR), as a proxy of hypovolemia, is associated with in-hospital death or admission to the ICU in a cohort of noncritically ill patients hospitalized with COVID-19 and hyponatremia detected at the time of hospital admission. We also investigated whether a rapid increase in serum sodium concentration during hospital stays affects clinical outcomes in hypovolemic patients with different baseline UCRs.

## 2. Materials and Methods

### 2.1. Study Design

This observational, retrospective, multicenter study involved three large hospitals in Northern Italy. Adult (≥18 years of age) patients diagnosed with COVID-19 and consecutively admitted to the three centers from January 2020 to May 2021 were included in the study. Clinical data were merged with the hospital lab database, and patients with at least one measurement of serum sodium, potassium, urea, and creatinine performed within 72 h since admission and with a serum sodium value below 135 mmol/L at the first measurement were included in the study. The patients lacking at least one valid measurement of serum sodium, potassium, urea, and creatinine within 72 h since admission were excluded.

Participants were followed up until the first occurrence of hospital discharge, transfer to another facility, or in-hospital death.

The study (STORM) was approved by the ethics committee of the coordinating center (San Gerardo Hospital, Monza) and by the IRB of each center and registered at ClinicalTrials.gov (accessed on 26 May 2023) (Identifier: NCT04670094, 15 December 2020).

### 2.2. Definition of Covariates

Study covariates included age, sex, history of comorbidities, drug treatments at admission, and blood chemistry parameters. We collected information on selected comorbidities, namely ischemic heart disease, heart failure, peripheral vascular disease, history of stroke, dementia, chronic obstructive pulmonary disease (COPD), liver failure, cancer, and diabetes mellitus. Charlson Comorbidity Index (CCI) was also computed.

Drug treatments at admission included angiotensin-converting enzyme inhibitors and angiotensin II receptor blockers classified as RAAS inhibitors, beta blockers, anticoagulants, and antiarrhythmic agents.

In addition to serum sodium, blood chemistry parameters included serum potassium, urea, and creatinine: hemoglobin, hematocrit, white blood cell count, and C-reactive protein.

Urea and creatinine measurements were used to compute UCR. Serum creatinine measurements were used to calculate the estimated glomerular filtration rate (eGFR) with the Chronic Kidney Disease Epidemiology Collaboration (CKD-EPI) equation [30]. Chronic kidney disease (CKD) was defined as eGFR < 60 mL/min/1.73 m^2^.

### 2.3. Outcomes

The primary outcome was all-cause mortality. The secondary outcome was admission to ICU during hospital stays.

### 2.4. Statistical Analysis

The study population was subdivided into two subgroups based on UCR value at admission. Patients with admission UCR equal to or higher than 40 were regarded as being hypovolemic. The UCR value of 40 is equivalent to the value of BUN/sCr ratio of 20 proposed as a clinical proxy of hypovolemia (26,13,14).

Continuous data were described using medians and quartiles (first-third quartile Q1–Q3) and compared using the Kruskal–Wallis rank test, while categorical data were described using counts and percentages and compared using the chi-square (χ^2^) test.

Estimated restricted cubic spline transformations were built using unadjusted logistic models on in-hospital mortality and admission to ICU to model the probability of death, as well as admission to ICU, by increasing level of admission UCR and placing five knots on the minimum, Q1, median, Q3, and maximum level of this ratio, respectively.

The Aalen–Johansen estimator was used to estimate the crude cumulative incidence of mortality and admission to ICU, accounting for discharge as a competing event, and the Gray test was used to test the null hypothesis of no difference in mortality and admission to ICU among the two groups according to admission UCR level.

Time-dependent Cox proportional hazards regression models were used to investigate the association between UCR and all-cause mortality, as well as admission to ICU. Urea-to-creatinine ratio during hospital stays was modeled as a time-varying covariate, and its association with outcomes was estimated for each 5-point increment. Potential confounders accounted for in the models were age, sex, CCI, treatment with diuretics or corticosteroids at any time during hospital stays, and serum potassium and eGFR at admission. We explored the impact of the time-varying serum UCR on in-hospital mortality patients treated with diuretics or corticosteroids at any time during hospital stays.

Furthermore, we also examined the prognostic role of changes in serum sodium values in hyponatremic patients when at least two serum sodium measurements were performed during the first 7 days of hospital stays. Increase in serum sodium values was categorized as lower or greater than 10 mmol/L. Logistic regression was applied separately in the two subgroups of hyponatremic patients classified according to UCR at admission, namely <40 and ≥40, to evaluate the prognostic impact of discrete changes in serum sodium within the first 7 days of hospital stays on in-hospital mortality.

Hazard ratios (HRs) or odds ratios (ORs) with 95% confidence intervals (CIs) were reported. SAS 9.4 was used for the statistical analyses, and the first-type error for tests was set at 0.05 (two-tailed).

## 3. Results

### 3.1. General Characteristics of Hyponatremic COVID-19 Patients

From an initial sample of 3470 patients admitted for COVID-19 between January 2020 and May 2021, we identified 258 patients who had at least one measurement of serum sodium, potassium, urea, and creatinine available within 72 h since hospital admission and had a serum sodium value below 135 mmol/L at the first measurement (Appendix A). Of these, 231 patients (89.5%) had more than one measurement of urea and creatinine during their hospital stay, and 199 patients (77.1%) had more than one measurement of serum sodium within one week of hospitalization.

Overall, the median (1st–3rd quartile) patient age was 69 (59–78) years, with a majority of males (69.0%). Among comorbidities, diabetes (31.9%) and CKD (35.9%) had the highest relative frequencies, followed by ischemic heart disease (17.5%). Treatment with RAAS inhibitors and beta-adrenergic blockers had a prevalence of 42.9% and 34.3%, respectively. The median (1st–3rd quartile) value of the serum UCR was 40.6 (31.1–51.1) (Table 1, overall sample).

Thus, we divided our patient population into two subgroups based on a serum UCR value of 40 and regarded the patients with a UCR ≥ 40 as being hypovolemic.

### 3.2. General Characteristics of Hyponatremic COVID-19 Patients Partitioned Based on a UCR < or ≥40 at Baseline

Patients with a UCR ≥ 40 were older and had a higher CCI than those with a UCR < 40. Specifically, the former group had a higher prevalence of ischemic heart disease and COPD (Table 1). Serum potassium values were slightly but significantly (*p* = 0.002) higher in the patients with a UCR ≥ 40, while serum creatinine and eGFR were not different compared to the patients with a UCR < 40 (*p* = 0.879 and *p* = 0.097, respectively). The patients with a UCR ≥ 40 at admission were also more likely to be on treatment with RAAS inhibitors and with beta-adrenergic blockers (Table 1).

### 3.3. Relationship of Serum Urea-to-Creatinine Ratio at Admission and the Incidence of In-Hospital Death or Admission to Intensive Care Unit in Hyponatremic COVID-19 Patients

We observed 52 deaths in this cohort; the crude probability of in-hospital death as a function of the UCR at admission is depicted in Figure 1.

It can be seen that the probability of in-hospital death increased linearly with the UCR until a value of approximately 40, leveling off after this threshold. This threshold corresponded to the median value of the UCR distribution in the whole population of COVID-19 patients with hyponatremia at hospital admission (Table 1). The crude probability of admission to the ICU (37 patients) as a function of the UCR at admission is shown in Appendix A. No specific trend was displayed in the probability of this outcome.

The crude cumulative incidence of in-hospital death was higher (*p* = 0.011) in patients with a UCR > 40 at baseline than in those with a UCR < 40 (Figure 2, panel A). On the other hand, the unadjusted cumulative incidence of ICU admission was similar (*p* = 0.516) in the two subgroups (Figure 2, panel B).

### 3.4. Relationship between Changes in the Urea/Creatinine Ratio and Outcomes in Hyponatremic COVID-19 Patients

We examined the prognostic impact of the UCR during hospital stays by modeling this index as a time-varying covariate. We observed that every 5-unit increase in the UCR was associated with an 8% increase in the hazard of all-cause death (HR = 1.08, 95% CI: 1.03–1.14, *p* = 0.001) after adjusting for potential confounders, including treatment with diuretics or corticosteroids at any time during hospital stays (Table 2, Model C). When these latter variables were not considered, the regressors’ results were very similar (Models A and B).

We also detected a 6% increase in the hazard of ICU admission associated with a 5-unit increase in the UCR, which, however, did not reach statistical significance (HR = 1.06, 95% CI: 0.99–1.12, *p* = 0.085) in the fully adjusted model (Table 3, Model C).

Patients with a baseline UCR ≥ 40 were more likely to undergo diuretic treatment during hospital stays than those who had a UCR < 40 (41/132, 31.3% vs. 21/126, 16.9%; *p* = 0.012). Similarly, the former were also more likely than the latter to undergo treatment with corticosteroids at any time during hospital stays (84/132, 64.1% vs. 55/126, 44.4%; *p* = 0.002). We explored the impact of a time-varying UCR on in-hospital mortality in hyponatremic patients who were or were not treated with diuretics and with corticosteroids (Appendix A) during hospital stays. We found that in patients who underwent diuretic treatment (Appendix A, Model A), a 5-unit increase in the UCR was associated with a 21% (HR = 1.21, 95% CI: 1.08–1.36, *p* = 0.001) greater hazard of all-cause death after adjusting for potential confounders. This did not hold true in the patients who were not treated with diuretics (Appendix A, Model B, HR = 1.05, 95% CI: 0.99–1.12, *p* = 0.095). In patients treated with corticosteroids (Appendix A, Model A), a 5-unit increase in the UCR was associated with a 12% (HR = 1.12, 95% CI: 1.05–1.19, *p* = 0.001) greater hazard of all-cause death after adjusting for potential confounders. This did not hold true in the patients who were not treated with corticosteroids (Appendix A, Model B, HR = 1.05 95% CI: 0.98–1.13, *p* = 0.179).

### 3.5. Prognostic Role of Changes in Serum Sodium Values in Hyponatremic Patients Partitioned Based on a UCR < or ≥40 at Baseline

Among the 199 (77.1%) hyponatremic patients with at least two measurements of sodium during the first 7 days from admission, 17.6% experienced an increase in serum sodium values equal to or greater than 10 mmol/L within 1 week of their hospital stay. The incidence of a serum sodium change of ≥10 mmol/L was higher in the subgroup of patients with a UCR > 40 compared to the subgroup with a UCR < 40 at admission (22/107, 20.6% vs. 13/92, 14.1%, Figure 3).

We, therefore, examined the prognostic impact of this increase (≥10 mmol/L) in serum sodium on in-hospital mortality in the two subgroups of hyponatremic patients with a UCR ≥ 40 or <40 at admission. In the subgroup of patients with a UCR ≥ 40 (Table 4, Model B), those with a variation ≥ 10 mmol/L of serum sodium had higher odds of in-hospital death (OR = 2.93, 95% CI: 1.03–8.36, *p* = 0.0443) compared to the patients who experienced a < 10 mmol/L change in serum sodium; this was not observed in patients with a UCR < 40 (Table 4, Model A).

## 4. Discussion

The main result of this study is that hyponatremic patients who developed hypovolemia during hospitalization for COVID-19 experienced an increase in the hazard of in-hospital death. Moreover, in hyponatremic patients with hypovolemia already present at admission, as evidenced by a UCR value ≥ 40 at baseline, a ≥10 mmol/L increase in serum sodium values within the first week of hospitalization was associated with a nearly three-fold increase in the odds of in-hospital death compared with patients with a UCR value < 40 at baseline.

The UCR has been used as an index of a contracted effective circulating volume and/or neurohormonal activation in several areas of clinical research, including ischemic stroke [24,26], heart failure [20,21], acute myocardial infarction [22,23], and subarachnoid hemorrhage [27]. In these pathological conditions, an elevated UCR value has also been reported as a predictor of poor outcomes.

Hyponatremia is a common finding in hospitalized patients, including those with COVID-19-associated pneumonia [8,9,10,11,12]. However, the reported prognostic impact of hyponatremia in this patient population, especially with respect to the risk of death, varies across published studies. Moreover, it is unclear whether a different state of extracellular volume repletion may affect the association of hyponatremia with clinical outcomes. Indeed, Tzoulis et al. [11] observed that serum urea values above 5 mmol/L, as a proxy of volume depletion, may be associated with higher mortality in hyponatremic COVID-19 patients. Moreover, Liu et al. [28] reported that a high BUN/Cr ratio value may be used as a prognostic tool in these patients, although those investigators did not stratify their population based on serum sodium values. Thus, our results extend the previous findings by Tzoulis et al. [11] and ourselves [12], suggesting that the condition of hypovolemia detected at admission in hyponatremic COVID-19 patients portends an adverse prognosis.

Diuretic treatment can induce neurohumoral activation due to volume depletion [18,31], thus increasing the UCR value. Indeed, a diuretic-induced increase in the UCR, associated with poor outcomes, was also observed in different patient populations, including patients with ischemic stroke [25,26] and congestive heart failure. In the latter population, excessive hemoconcentration, expressed by a large increase in the BUN/Cr ratio value in patients treated with high-dose loop diuretics, was associated with increased mortality [20,32]. Furthermore, in patients with acute decompensated heart failure, the combination of hyponatremia (i.e., serum sodium values < 136 mmol/L) and elevated BUN values (i.e., >21 mg/dL) at discharge was associated with a higher hazard of death compared with hyponatremia alone or elevated BUN alone [33]. These findings support the role of excessive neurohormonal activation in predicting adverse prognosis. In our population, approximately one third of all hyponatremic patients who developed hypovolemia were treated with diuretics during their hospital stays. Therefore, we examined the association of a time-varying UCR with outcomes by adjusting for diuretic treatment administered at any time during hospital stays. Moreover, we also performed a stratified analysis of hyponatremic patients who were or were not treated with diuretics and found that increasing values of the UCR remained independently associated with an increased hazard of death in the former but not in the latter. Thus, we hypothesize that in our cohort of patients admitted with COVID-19 and hyponatremia at baseline, the adverse prognostic impact of the UCR increasing during hospital stays could have been related to neurohormonal hyperactivation secondary to worsening volume depletion, particularly in those submitted to diuretic treatment.

Hypovolemic patients with symptomatic hyponatremia, especially those treated with thiazides, are more prone to a rapid and greater increase in serum sodium concentration due to rapid volume expansion during treatment with hypertonic saline compared to normovolemic patients, conceivably due to the appropriate suppression of vasopressin secretion [34]. The overcorrection of serum sodium values may also be induced by infusions of isotonic saline solution in hypovolemic patients with hyponatremia [8]. In hyponatremic patients, the overcorrection of serum sodium values may be associated with severe brain damage and poor prognosis [35]. In our cohort of hyponatremic COVID-19 patients, we observed that the incidence of a change in serum sodium equal to or greater than 10 mmol/L within 1 week of hospital stays was higher in the subgroup of those who were hypovolemic (i.e., who had a UCR ≥ 40) at the time of the hospital admission compared to the subgroup who were not hypovolemic. Moreover, in hypovolemic patients, a ≥10 mmol/L increase in serum sodium within one week since admission was associated with higher odds of in-hospital death compared with nonhypovolemic patients experiencing a comparable change in serum sodium values. Our results are in line with those of Tzoulis et al. [11], who observed the highest mortality rate in COVID-19 patients experiencing a ≥8 mmol/L increase in serum sodium values in the first 5 days of hospital stays. We observed a ≥10 mmol/L increase in serum sodium in a relatively large fraction (35/199, 17.6%) of our hyponatremic patients, in whom at least two measurements of serum sodium were performed within one week since admission. Indeed, in patients admitted for COVID-19, hypovolemia could develop due to vomiting, diarrhea, or diuretic treatment. While we did not have information on the type and amount of fluids administered to our patients during hospital stays, we speculate that in a fraction of these patients, volume replenishment could have induced a diuresis-driven overcorrection of serum sodium values, which could have impacted overall survival.

The main strength of this study is that it was conducted on a representative sample of COVID-19 patients who had hyponatremia at hospital admission, from whom several measurements of serum urea, creatinine, and sodium values were collected during hospital stays. This allowed us to model the UCR as a time-varying variable and to explore the relative impact of worsening hypovolemia and serum sodium changes on clinical outcomes. Moreover, we collected information on treatment with diuretics and corticosteroids during hospital stays, which allowed us to analyze the interaction of the UCR with these treatments and perform a subgroup analysis in the patients who did or did not receive diuretics during hospitalization.

Several limitations of our study must also be acknowledged. Firstly, while the UCR may be regarded as an index of contracted effective circulating volume [16,17,18], it is also affected by protein intake, gastrointestinal bleeding, muscle mass, and accelerated catabolism, especially in critically ill patients [36]. However, the patients enrolled in our study were not critically ill at baseline, nor did an increasing UCR predict admission to the ICU. Treatment with glucocorticoids may also increase the UCR by promoting protein breakdown [37]. However, although a greater fraction of the patients with a UCR value ≥ 40 received treatment with glucocorticoids compared to the patients who maintained a UCR < 40, we did not detect a significant interaction between the UCR value and corticosteroid treatment while analyzing the association of the time-varying UCR with in-hospital mortality. Secondly, we did not collect information on changes in the acid-base status; thus, we could not explore whether a putative increase in plasma bicarbonate concentration may have paralleled the increase in the UCR value, thus reinforcing the interpretation of a UCR ≥ 40 as an index of hypovolemia in hyponatremic patients, especially in those who received diuretics during hospital stays. However, patients with diarrhea-induced volume depletion usually develop metabolic acidosis rather than metabolic alkalosis. Hence, the impact of hypovolemia on changes in the acid-base status may be confounded by different clinical situations and may limit the clinical usefulness of those changes in the assessment of effective circulating volume. Thirdly, we had no information on urine output and fluid administration during hospital stays; thus, we could not analyze the impact of fluid balance on the changes in the UCR value and serum sodium. However, we observed that a greater fraction of hyponatremic patients who developed a UCR value ≥ 40 received diuretics during hospitalization compared with those in whom the UCR remained < 40, which suggests that diuretics may have contributed to fluid losses and may have favored a greater increase in serum sodium in the former. Finally, as this was an observational study, we cannot infer a cause–effect relationship of the changes in the UCR and serum sodium with patient outcomes. Notwithstanding adjustments for comorbidity burden, kidney function at baseline, and treatment with diuretics or corticosteroids during hospitalization in all analyses, residual confounding is possible, and our results must be considered merely hypothesis-generating.

## 5. Conclusions

Our results suggest that hypovolemia occurring during hospital stays in noncritically ill COVID-19 patients with hyponatremia detected at hospital admission may have an adverse prognostic impact. Moreover, in hypovolemic patients with hyponatremia, a ≥10 mmol/L increase in serum sodium may further worsen in-hospital survival.

Our findings reinforce and extend the existing data in the literature suggesting that hypovolemia developing in COVID-19 hospitalized patients with hyponatremia is associated with unfavorable outcomes. These findings may not necessarily be valid only in COVID-19 patients, as previous research indicates that an increasing UCR bears prognostic information in different patient populations. Thus, we suggest that clinicians should monitor the UCR in hospitalized patients with hyponatremia to detect those at the highest risk and take appropriate action to reverse hypovolemia using judicious volume expansion. However, while restoring an adequate effective circulating volume, great caution should be exerted to avoid a rapid and/or excessive increase in serum sodium concentration.

## Figures and Tables

**Figure 1 biomedicines-11-01555-f001:**
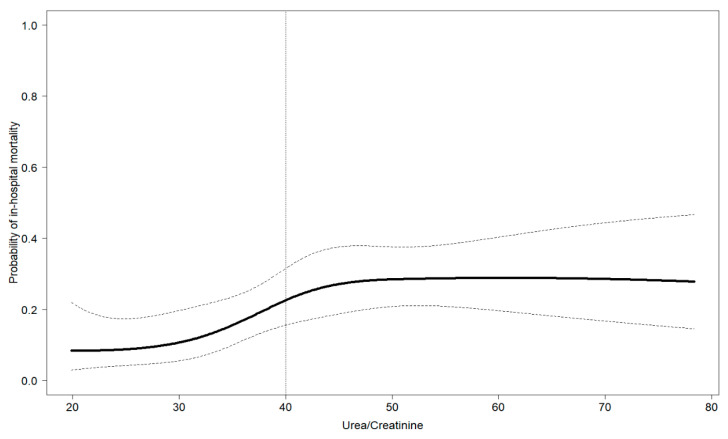
Estimated restricted cubic spline transformation of the probability of in-hospital mortality by values of the urea-to-creatinine ratio at admission. Estimate from unadjusted logistic model on in-hospital mortality. Restricted cubic spline with five knots on the minimum, first, second, and third quartile, and maximum value of the admission urea/creatinine ratio.

**Figure 2 biomedicines-11-01555-f002:**
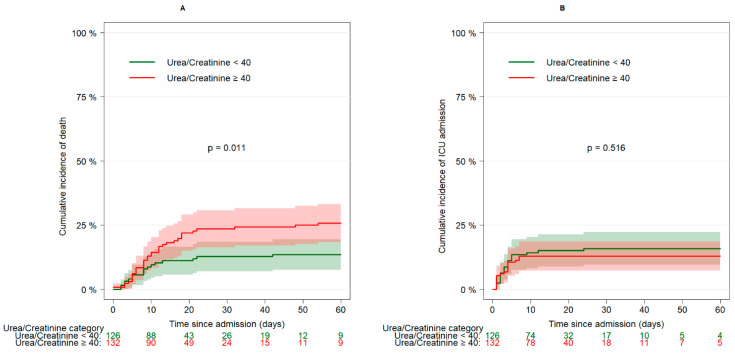
Cumulative incidence of death (panel **A**) and ICU admission (panel **B**) by categories of urea-to-creatinine ratio at admission.

**Figure 3 biomedicines-11-01555-f003:**
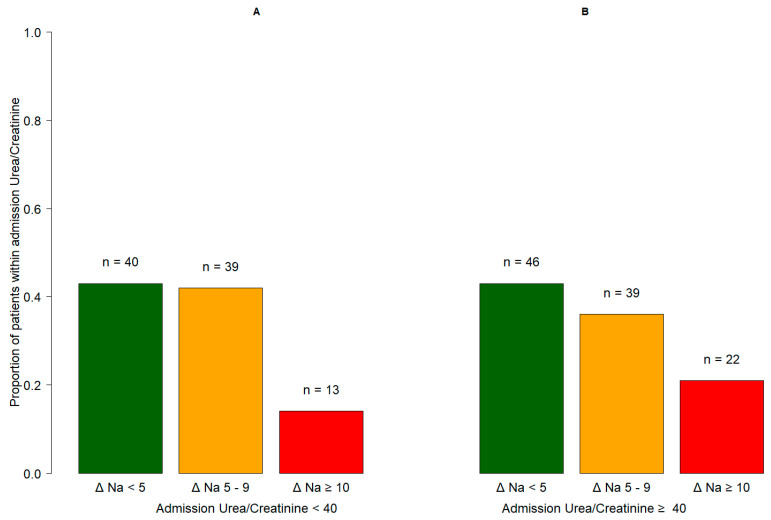
Proportion of patients’ maximum variation of sodium (ΔNa, mmol/L) within 7 days after admission in patients with baseline urea-to-creatinine ratio <40 (panel **A**, *n* = 92) and ≥40 (panel **B**, *n* = 107). Only patients with at least two measurements of sodium within 7 days after admission were included in the analysis.

**Table 1 biomedicines-11-01555-t001:** Baseline characteristics of study population overall and within two strata defined according to categories of urea-to-creatinine ratio at admission. CKD, chronic kidney disease; COPD, chronic obstructive pulmonary disease; eGFR, estimated glomerular filtration rate; RAAS, renin–angiotensin–aldosterone system.

	Overall	Urea/Creatinine < 40	Urea/Creatinine ≥ 40	*p*	Missing (%)
*n*	258	126	132		
Male (*n*, %)	178 (69.0)	89 (70.6)	89 (67.4)	0.673	0
Age (years, median [Q1–Q3])	69 [59, 78]	65 [52, 75]	71 [65, 80]	<0.001	0
Hematocrit (%, median [Q1–Q3])	37.8 [34.0, 41.0]	38.2 [34.3, 41.5]	36.9 [34.0, 40.6]	0.432	2.7
Hemoglobin (g/dL, median [Q1–Q3])	12.8 [11.3, 13.9]	12.9 [11.3, 13.9]	12.5 [11.2, 14.0]	0.477	2.7
White blood cell count (10^3^/µL, median [Q1–Q3])	6.38 [4.75, 10.10]	6.00 [4.45, 9.60]	7.00 [4.84, 10.44]	0.213	3.9
Urea (mg/dL, median [Q1–Q3])	41.5 [30.0, 63.8]	31.0 [24.0, 43.7]	52.0 [40.0, 78.7]	<0.001	0
Creatinine (mg/dL, median [Q1–Q3])	1.00 [0.82, 1.39]	1.00 [0.82, 1.33]	1.01 [0.82, 1.41]	0.879	0
Urea/Creatinine (median [Q1–Q3])	40.6 [31.1, 51.1]	31.0 [25.9, 34.8]	50.5 [44.5, 59.4]	-	0
Sodium (mmol/L, median [Q1–Q3])	133 [131, 134]	133 [131, 134]	133 [131, 134]	0.412	0.4
Potassium (mmol/L, median [Q1–Q3])	4.11 [3.76, 4.57]	4.02 [3.66, 4.38]	4.22 [3.92, 4.70]	0.002	0.4
C-reactive protein (mg/L, median [Q1–Q3])	84.6 [41.2, 133.0]	86.3 [46.5, 132.2]	81.4 [40.6, 133.0]	0.721	9.3
Ischemic heart disease (*n*, %)	45 (17.5)	14 (11.2)	31 (23.5)	0.015	0.4
Heart failure (*n*, %)	14 (5.4)	5 (4.0)	9 (6.8)	0.472	0.4
Peripheral vascular disease (*n*, %)	24 (9.3)	7 (5.6)	17 (12.9)	0.073	0.4
History of stroke (*n*, %)	24 (9.3)	11 (8.8)	13 (9.8)	0.941	0.4
Dementia (*n*, %)	13 (5.1)	6 (4.8)	7 (5.3)	1	0.4
COPD (*n*, %)	26 (10.1)	4 (3.2)	22 (16.7)	0.001	0.4
Liver failure (*n*, %)	19 (7.4)	6 (4.8)	13 (9.8)	0.191	0.4
eGFR (mL/min/1.73 m^2^, median [Q1–Q3])	69.7 [47.3, 87.8]	72.9 [51.6, 89.3]	67.1 [44.2, 84.5]	0.097	2.7
CKD (eGFR < 60 mL/min/1.73 m^2^, *n*, %)	90 (35.9)	39 (31.5)	51 (40.2)	0.191	2.7
Cancer (*n*, %)	35 (13.8)	18 (14.5)	17 (13.1)	0.88	1.6
Diabetes mellitus (*n*, %)	82 (31.9)	33 (26.4)	49 (37.1)	0.087	0.4
Charlson Comorbidity Index (median [Q1–Q3])	2 [0, 3]	1 [0, 3]	2 [1, 3]	0.001	0
Systolic blood pressure (mmHg, median [Q1–Q3])	130 [120, 145]	130 [120, 147]	130 [120, 145]	0.78	2.7
Diastolic blood pressure (mmHg, median [Q1–Q3])	75 [67, 80]	75 [67, 80]	75 [67, 80]	0.89	2.7
Heart rate (bpm, median [Q1–Q3])	88 [75, 102]	87 [75, 100]	89 [75, 105]	0.259	33.7
RAAS inhibitors (*n*, %)	102 (42.9)	40 (33.9)	62 (51.7)	0.008	7.8
Beta blockers (*n*, %)	86 (34.3)	31 (25.4)	55 (42.6)	0.006	2.7
Anticoagulants (*n*, %)	28 (11.0)	12 (9.8)	16 (12.2)	0.671	1.6
Antiarrhythmics (*n*, %)	11 (4.3)	4 (3.3)	7 (5.4)	0.601	1.9

**Table 2 biomedicines-11-01555-t002:** Multivariable Cox regression models on mortality by time-varying urea-to-creatinine ratio. CI, confidence interval; eGFR, estimated glomerular filtration rate; HR, hazard ratio.

	Model A (*n* = 258, Deaths = 52)		Model B (*n* = 255, Deaths = 52)		Model C (*n* = 247, Deaths = 49)	
Parameter	HR (95% CI)	*p* Value	HR (95% CI)	*p* Value	HR (95% CI)	*p* Value
5-point increment of urea/creatinine	1.06 (1.02–1.12)	0.0093	1.07 (1.02–1.12)	0.0082	1.08 (1.03–1.14)	0.0011
Age (years)	1.04 (1.01–1.07)	0.0109	1.04 (1.01–1.07)	0.0131	1.03 (1.01–1.06)	0.0187
Male (yes vs. no)	1.40 (0.75–2.62)	0.2913	1.41 (0.75–2.63)	0.2884	1.68 (0.87–3.23)	0.1220
Charlson Comorbidity Index	1.46 (1.24–1.71)	<0.001	1.46 (1.23–1.74)	<0.001	1.18 (0.95–1.46)	0.1471
Diuretic during hospitalization (yes vs. no)			0.97 (0.52–1.80)	0.9246	1.12 (0.59–2.12)	0.7368
Corticosteroid during hospitalization (yes vs. no)			0.90 (0.51–1.59)	0.7216	0.53 (0.28–1.01)	0.0519
Potassium at admission (mmol/L)					1.61 (1.04–2.48)	0.0346
eGFR at admission (mL/min/1.73 m^2^)					0.97 (0.96–0.99)	0.0007

**Table 3 biomedicines-11-01555-t003:** Multivariable Cox regression models on ICU admission by time-varying urea-to-creatinine ratio. CI, confidence interval; eGFR, estimated glomerular filtration rate; HR, hazard ratio.

	Model A (*n* = 258, ICU = 37)		Model B (*n* = 255, ICU = 37)		Model C (*n* = 247, ICU = 37)	
Parameter	HR (95% CI)	*p* Value	HR (95% CI)	*p* Value	HR (95% CI)	*p* Value
5-point increment of urea/creatinine	1.04 (0.98–1.11)	0.1843	1.06 (0.99–1.12)	0.0664	1.06 (0.99–1.12)	0.0849
Age (years)	0.97 (0.94–0.99)	0.0112	0.96 (0.94–0.99)	0.0051	0.97 (0.94–0.99)	0.0108
Male (yes vs. no)	2.29 (0.95–5.51)	0.0652	2.28 (0.95–5.49)	0.0657	2.19 (0.90–5.36)	0.0833
Charlson Comorbidity Index	0.96 (0.76–1.21)	0.7272	0.96 (0.76–1.22)	0.7586	1.07 (0.81–1.41)	0.6273
Diuretic during hospitalization (yes vs. no)			0.93 (0.38–2.27)	0.8754	0.97 (0.39–2.38)	0.9382
Corticosteroid during hospitalization (yes vs. no)			0.63 (0.32–1.23)	0.1746	0.73 (0.36–1.48)	0.3814
Potassium at admission (mmol/L)					0.66 (0.36–1.22)	0.1848
eGFR at admission (mL/min/1.73 m^2^)					1.01 (0.99–1.03)	0.3601

**Table 4 biomedicines-11-01555-t004:** Multivariable logistic regression models on mortality by maximum variation of sodium within 7 days after admission, stratified by value of the urea-to-creatinine ratio at admission (only the patients with at least two measurements of sodium within 7 days after admission were included in the analysis). CI, confidence interval; OR, odds ratio.

	Urea/Creatinine < 40		Urea/Creatinine ≥ 40	
	Model A (*n* = 92, Deaths = 13)		Model B (*n* = 107, Deaths = 30)	
Parameter	OR (95% CI)	*p* Value	OR (95% CI)	*p* Value
ΔNa ≥ 10 mmol/L vs. ΔNa < 10 mmol/L	0.22 (0.01–3.37)	0.2766	2.93 (1.03–8.36)	0.0443
Age (years)	1.05 (0.99–1.12)	0.1241	1.04 (0.99–1.09)	0.1144
Male (yes vs. no)	4.57 (0.77–27.1)	0.0936	0.79 (0.28–2.19)	0.6474
Charlson Comorbidity Index	1.79 (1.09–2.92)	0.0201	1.32 (1.02–1.71)	0.0335

## Data Availability

Please contact the corresponding author regarding data requests.

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
