# Peer review of "Elevated Serum Urea-to-Creatinine Ratio and In-Hospital Death in Patients with Hyponatremia Hospitalized for COVID-19"

_biomedicines, 2023, doi:10.3390/biomedicines11061555_

Round 1

Reviewer 1 Report

For authors,

The authors conducted exciting new research on sodium imbalances, which are common among hospitalized patients with COVID-19 and can lead to life-threatening complications if not managed properly. Hyponatremia is the most frequently observed electrolyte disturbance in clinical practice, especially in patients with COVID-19, and is associated with an increased risk of death.

Major recommendations:

1. Several mechanisms are responsible for causing hyponatremia in patients with COVID-19. For a better understanding, I suggest you to detail the mechanisms underlying the occurrence of hyponatremia:

- common mechanism is Inadequate Antidiuretic Hormone Syndrome (SIADH);

- the likely mechanism involved would be the reduction of dietary sodium intake together with gastrointestinal sodium losses (vomiting and diarrhea);

- another mechanism that could have played a role could be the dysfunction of the renin-angiotensin-aldosterone system (RAAS);

- the administration of diuretics in volume overloaded COVID-19 patients could be a fourth mechanism for the development of hyponatremia through increased urinary sodium excretion.

2. Pay attention to all abbreviations and their meanings, as they must be entered as they first appear in the text; check the manuscript.

3. I suggest you improve the conclusions and emphasize what the current research adds to the existing research - make some arguments/strong points in the conclusions.

4. Please check that the resolution of the figures is in accordance with the guidelines of the journal (high enough resolution of at least 1000 pixels width/height or a resolution of 300 dpi or higher).

Sincerely yours,

Minor editing of English language required

Author Response

Reviewer 1: Please look at the uploded file

Reviewer 2 Report

The manuscript entitled Elevated serum urea to creatinine ratio and in-hospital death in patients with hyponatremia hospitalized for COVID-19 is an original article well written.

This retrospective study has a methodology that could be discussable.

What were the exclusion criteria? There are some drugs, which have hyponatremia as adverse effect. It is mandatory to specify as exclusion criteria these drugs, which could be frequently in older patients (example psychiatric drugs). This could be an important issue of the methodology. Please clearly specify all potentially causes of hyponatremia that must be excluded in these patients.

The results are well presented. The discussions might be synthetized to be easier readable.

The manuscript entitled Elevated serum urea to creatinine ratio and in-hospital death in patients with hyponatremia hospitalized for COVID-19 is an original article well written.

This retrospective study has a methodology that could be discussable.

What were the exclusion criteria? There are some drugs, which have hyponatremia as adverse effect. It is mandatory to specify as exclusion criteria these drugs, which could be frequently in older patients (example psychiatric drugs). This could be an important issue of the methodology. Please clearly specify all potentially causes of hyponatremia that must be excluded in these patients.

The results are well presented. The discussions might be synthetized to be easier readable.

Author Response

Reviewer 2: Please look at the uploded file

Reviewer 3 Report

Regolisti et al. examined the relationship between elevated serum urea to creatinine ratio and in-hospital death in patients with hyponatremia and COVID-19.  They showed that hyponatremia was associated with increased mortality in COVID-19 patients.  In addition, they showed that a > 10 mmol/l increase in serum sodium within the first week of hospitalization further worsened the prognosis. 

Comments:

1. The study is retrospective.  It is strange to use the word “primary end-point”. 

2. Continuous data were described by medians and Q1-Q3.  However, if variables were distributed normally, they should be expressed as means +/- SD, and statistical analysis be performed appropriately. 

The manuscript is well written.

Author Response

Reviewer 3: Please look at the uploded file

Round 2

Reviewer 1 Report

Dear Authors,

I appreciate your interest in the suggestions made, and I hope I've helped improve your article.

Reviewer 2 Report

Thank you for responding to my comments.